# Three new species of *Acalypha* L. (Euphorbiaceae, Acalyphoideae) from Tanzania and Angola and their conservation status

Iris Montero-Muñoz[1,2], José María Cardiel[3,4], Lucía Villaescusa-González[5], Pablo Muñoz-Rodríguez[5*]

1 Universidad de Alcalá, Faculty of Sciences, Department of Life Sciences, Madrid, Spain, 2 Real Jardín Botánico, CSIC, Madrid, Spain, 3 Universidad Autónoma de Madrid, Faculty of Sciences, Department of Biology, Madrid, Spain, 4 Centro de Investigación en Biodiversidad y Cambio Global (CIBC-UAM), Universidad Autónoma de Madrid, Madrid, Spain, 5 Universidad Complutense de Madrid, Faculty of Biological Sciences, Department of Biodiversity, Ecology and Evolution, Madrid, Spain

* pablo.munoz@ucm.es

## Abstract

We describe *Acalypha brevipetiolata* and *A. bracteolata* from Tanzania, and *A. linearis* from Angola, as species new to science. We provide illustrations, distribution maps, and preliminary conservation assessments. We discuss their significance in the context of both countries and within the framework of an ongoing taxonomic monograph of the genus in Africa. These are the first new *Acalypha* species described in Africa in nearly two decades, in Tanzania in five decades, and in Angola in over a century.

## 1. Introduction

*Acalypha* L. (Euphorbiaceae Juss.) is a large plant genus with over 450 accepted species and a mainly pantropical distribution. As with many other tropical plant groups, knowledge of the genus remains uneven. *Acalypha* has been extensively studied in South America, where the authors of this paper have worked for the past three decades [1,2], and significant taxonomic revisions have also been conducted in the Western Indian Ocean Region, including Madagascar [3], and in Southeast Asia [4]. In contrast, Africa remains one of the least well-documented regions for the genus.

Few studies have focused on *Acalypha* in Tanzania or Angola, the two countries where the new species reported here were discovered. In Tanzania, the most comprehensive floristic treatment remains Radcliffe-Smith's work for the *Flora of Tropical East Africa* [5]. The only other taxonomic work we are aware of is the PhD thesis of R.L.M. Mahunnah, which questioned species boundaries within *Acalypha* but did not propose taxonomic rearrangements [6].

**Data availability statement:** The DNA sequence is submitted to GenBank under accession number PV368855. The remaining supporting data are within the paper and its Supporting information files.

**Funding:** The authors are grateful for the financial support provided by the research project PID2022-139634NA-I00M funded by Ministerio de Ciencia, Innovación y Universidades, MCIN/AEI/10.13039/501100011033/FEDER, UE. PMR was supported by a Ramón y Cajal research contract, grant ref. RYC2021-032489-I, funded by Ministerio de Ciencia, Innovación y Universidades, and IMM by a Juan de la Cierva – Formación-2021, grant ref. FJC2021-046607-I, funded by Ministerio de Ciencia, Innovación y Universidades, MCIN/AEI/10.13039/501100011033 and the European Union NextGenerationEU/PRTR during the work on this project. There was no additional external funding received for this study.

**Competing interests:** The authors have declared that no competing interests exist.

The situation in Angola is even more precarious: the only available reference about *Acalypha* is a checklist lacking taxonomic information [7]. Online herbarium records for Angolan *Acalypha* are extremely limited (e.g., fewer than 300 records in GBIF), particularly for the region where our newly described Angolan species was collected, making broad taxonomic comparisons difficult. Although Tanzania has more digitised records, coverage remains sparse compared to other African countries.

In a recently published nomenclatural review [8] we recognised 70 native species in mainland Africa, with 28 species recorded from Tanzania and 23 from Angola. Most species have not been critically examined or had their boundaries tested since their original descriptions —often over a century ago—, and a significant proportion (ca. 20%) are known only from type specimens or a handful of collections. The scarcity of data hampers any study on this megadiverse group of plants, emphasising the need for urgent taxonomic revision.

To address these gaps, in 2023 we started work on a continental-scale taxonomic monograph of *Acalypha* in mainland Africa. Here, we formally describe three species new to science discovered during our study of herbarium material, integrating morphological and, where possible, molecular data. We also discuss their phylogenetic and biogeographical significance in the context of *Acalypha* diversity in Africa.

## 2. Materials and methods

### 2.1. Taxonomic approach

We based our taxonomic decisions primarily on morphological evidence, complemented by geographical and ecological data. For one species (*Acalypha brevipetiolata*), we also obtained high-quality DNA for sequencing using the Sanger method (see below). We derived morphological descriptions and illustrations from herbarium specimens housed in BR, K, L, LISC, LISU, MA, MO, P and WAG (abbreviations according to [9]). We examined all cited specimens and provide herbarium identifiers when available.

To conduct comparative morphological analyses, we examined herbarium specimens of other African *Acalypha* species. We extracted information on habit, plant size, and habitat from field notes recorded on specimen labels. We followed a heuristic approach to delimit the new species [10], which allows us to recognise species boundaries based on consistent but not necessarily absolute discontinuities, ensuring taxonomic progress while accommodating potential future refinements. It also avoids rigid criteria that may lead to over-splitting or over-lumping, making it particularly useful for taxa with limited molecular data, as is the case here.

We maintain all taxonomic and biogeographical data on *Acalypha* in the regularly updated *Acalypha Taxonomic Information System*, available at https://acalypha.es.

### 2.2. Nomenclature

The electronic version of this article in Portable Document Format (PDF) in a work with an ISSN or ISBN will represent a published work according to the International Code of Nomenclature for algae, fungi, and plants, and hence the new names contained in the electronic publication of a PLOS ONE article are effectively published under that Code from the electronic edition alone, so there is no longer any need to provide printed copies.

In addition, new names contained in this work have been submitted to IPNI, from where they will be made available to the Global Names Index. The IPNI LSIDs can be resolved and the associated information viewed through any standard web browser by appending the LSID contained in this publication to the prefix http://ipni.org/. The online version of this work is archived and available from the following digital repositories: PubMed Central, LOCKSS, and Docta Complutense.

## 2.3. DNA extraction, sequencing and phylogenetic analysis

We attempted DNA extraction and amplification from multiple collections of the new species to place them in a phylogenetic context. However, despite several attempts using different methods, we obtained high-quality DNA sequences for only one specimen of *A. brevipetiolata* (*S. Bigdoog 8090*). Future efforts, possibly using fresher material, may improve sequencing success for the other taxa. Following previous studies [4,11] and our ongoing work on *Acalypha* (unpublished), we targeted the nuclear Internal Transcribed Spacer (*nrITS*) DNA region.

We extracted DNA from herbarium specimens using the CTAB method [12] with modifications (S1 Table) and quantified it using Qubit™ dsDNA BR Standard (Invitrogen). Due to DNA degradation, we amplified the *nrITS* region in two parts using primers *ITSw1f-ITSp2r* (5'-CCTTATCATTTAGAGGAAGGAG; 5'-GCCRAGATATCCGTTGCCGAG) [13] and *ITSp3f-ITSw2r* (5'-YGACTCTCGGCAACGGATA; 5'-TATGCTTAAAYTCAGCGGGT) [14].

We performed PCR reactions in a 25 μl volume, containing 12 μL of MyTaq Red Mix (Bioline), 10 μL $H_2O$, 1 μL of each primer (10μM), and 1 μL of genomic DNA, and standard PCR conditions: initial denaturation at 95°C for 5 minutes, followed by 36 cycles of 95ºC for 1 minute, 50ºC for 1.5 minutes, and 72ºC for 45 seconds, with a final extension at 72°C for 10 minutes. We purified the PCR products using ExoSap and sent them for sequencing at MACROGEN (Macrogen, Madrid, Spain) with the same amplification primers. We deposited the newly generated sequence in GenBank under accession number PV368855.

To infer phylogenetic relationships, we included 144 *nrITS* sequences previously generated by [11] and available on GenBank to represent *Acalypha* diversity, using *Bernardia* Houst. ex Mill., *Erythrococca* Benth., *Mareya* Baill. and *Micrococca* Benth. (one species each) as outgroups.

We first used NCBI BLAST with default settings to confirm sequence quality. We aligned the DNA sequences using MAFFT v.7.310 (*--auto --adjustdirection*) [15] and refined the alignment with trimAL (*--automated1*) [16]. We inferred a Maximum Likelihood phylogeny with IQ-TREE 2.4.0 (random seed number 524112, one thread) [17], automatic model selection with ModelFinder (SYM+I+G4 based on the Bayesian Information Criterion) [18] and 1,000 ultrafast bootstrap replicates. In the resulting phylogenies, we collapsed all nodes with bootstrap support below 60% into polytomies. All phylogenetic analysis files are available in S1 File.

## 2.3. Distribution maps and conservation assessments

To visualise the geographical distribution of the new species, we generated distribution maps using QGIS Desktop 3.28.4 and CGRI layers [19]. We conducted conservation assessments following the IUCN Red List Categories and Criteria [20]. We calculated the Area of Occupancy (AOO) and Extent of Occurrence (EOO) using GeoCAT [21], applying the standard 2×2 km grid cell size recommended by IUCN. We have not yet completed a formal IUCN assessment.

## 3. Results and discussion

### 3.1. New species of *Acalypha*

#### 3.1.1. *Acalypha brevipetiolata*. *Acalypha brevipetiolata* I.Montero, Cardiel & P.Muñoz, *sp. nov*. [urn:lsid:ipni.org: names:77368649-1] Type: Tanzania, T4, Mpanda District: 3 km S of Uzondo Camp, 1600 m, 5º31'S 30º32'E, 11 Mar 2009. *S. Bidgood, G. Leliyo & K. Vollesen 809*0 (MO [MO-3888236], holo.; BR [BR0000016110199], K, WAG [WAG.1196761], P [P00887479], iso.). Fig 1.

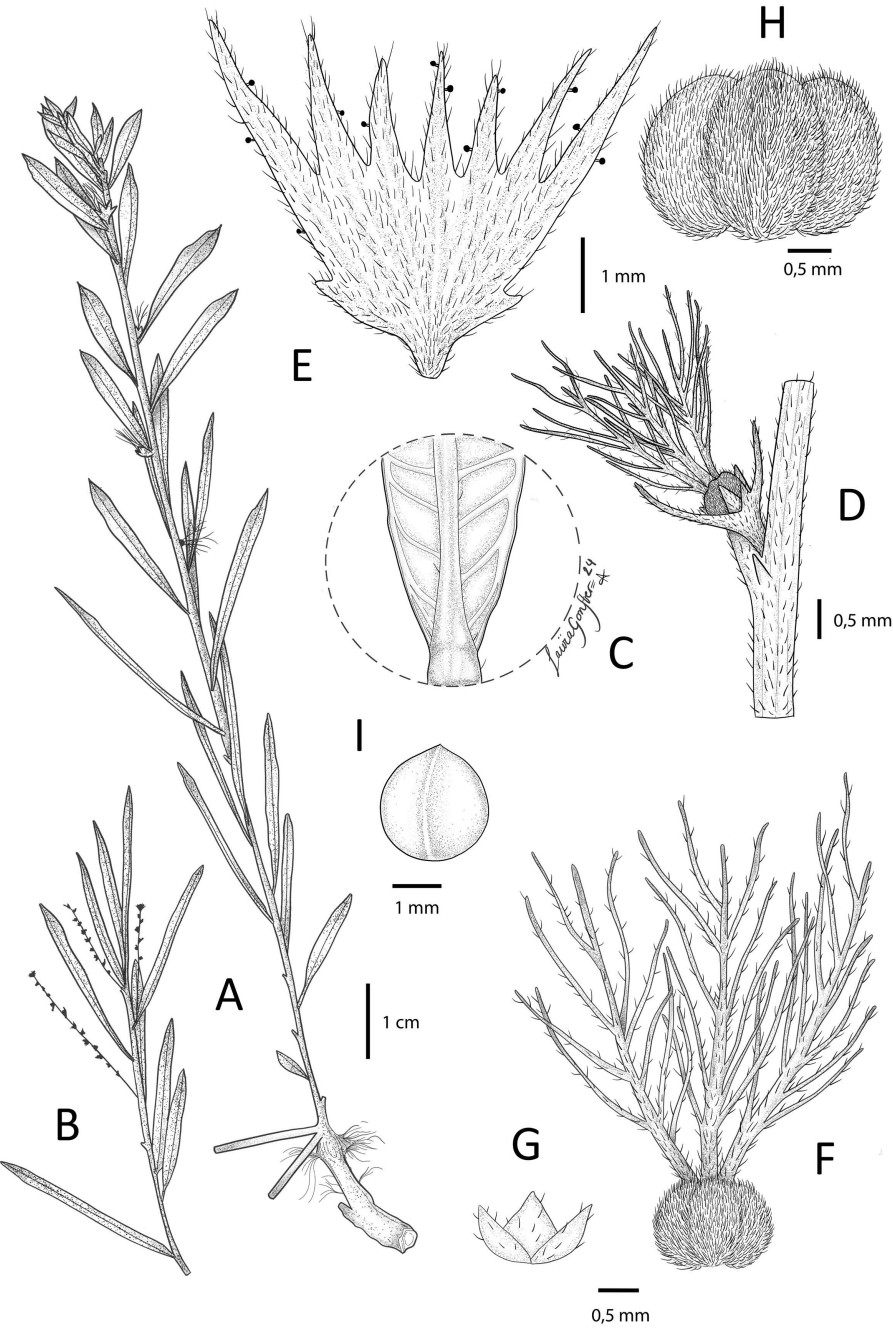

**Fig 1. *Acalypha brevipetiolata* I. Montero, Cardiel & P.Muñoz.** A. Habit of female branch. B. Habit of male branch. C. Detail of lower leaf surface. D. Detail of node, stipules, and female flower. E. Mature female bract. F. Ovary and styles. G. Calyx of the female flower. H. Capsule. I. Seed. Based on *S. Bidgood, G. Leliyo & K. Vollesen 8090*. Illustration by Laura González Hernández.

*Diagnosis*. *Acalypha brevipetiolata* is primarily characterised by its reddish branches, very short petioles (1−3 mm), linear to linear-lanceolate leaf blades with entire to sparsely denticulate margins, and female inflorescences with a solitary, shortly pedunculated bract with an irregularly dentate margin (8–9 teeth) and with both simple and glandular trichomes.

*Description.* Perennial herb up to 25 cm tall, monoecious, with numerous erect to decumbent stems from a large woody rootstock. Branches reddish, pubescent with short, appressed trichomes, glabrescent when mature. Axillary buds inconspicuous. Stipules inconspicuous, promptly deciduous, up to 0.5 mm long, triangular, acuminate, sparsely hairy with simple, minute trichomes. Petioles 1−3 mm long, more or less flattened at base, subglabrous with some short, simple trichomes. Leaf blades (1.5−)2.5−4(−4.5) × (0.15−)0.2−0.4 cm, linear to linear-lanceolate, chartaceous; base acute to slightly decurrent; apex acute, and sometimes mucronate or slightly callose; margins entire to sparsely denticulate; upper surface glabrous; lower surface subglabrous with some appressed, simple trichomes on veins; venation pinnate, secondary veins 8–10 per side, prominent on lower surface. Inflorescences unisexual, axillary, in different stems. Male inflorescences up to 2.6 cm long; peduncle up to 5 mm long, pubescent with short, simple trichomes; flowers sparsely glomerate; bracts minute up to 0.8 mm long, ovate-lanceolate, glabrous. Female inflorescences with a solitary bract, shortly pedunculate, peduncle up to 0.7 mm long, bracts enlarging in fruit to c. 5 × 5 mm, pubescent with short, simple, appressed trichomes and some glandular trichomes at margin; margin irregularly dentate, teeth 8–9, triangular-lanceolate, acute, ½–⅓ of the total bract length, two of them smaller basal teeth; bracteoles absent. Male flowers subsessile, pedicel up to 0.3 mm long, glabrous; flower buds not seen (fallen). Female flowers one per bract, sessile; sepals 3, up to 0.7 mm long, distinct, triangular-lanceolate, sparsely hairy, with short, simple trichomes; ovary c. 1 mm diameter, 3-lobed, densely pubescent with short, simple trichomes; styles 3, up to 3 mm long, distinct at base, with conspicuous central axis, each divided into 8–10 long, slender branches, sparsely hairy with some short, simple trichomes. Capsules up to 3 mm diameter, surface smooth, densely pubescent with short, simple trichomes. Seeds c. 2.1 mm diameter, globose, smooth.

*Evolutionary relationships.* *Acalypha brevipetiolata* belongs to a highly supported clade including *A. cupricola* Robyns ex G.A.Levin, *A. manniana* Müll.Arg., *A. polymorpha* Müll.Arg., and *A. welwitschiana* Müll.Arg. (Fig 2 and S1 Fig).

*Distribution and habitat.* *Acalypha brevipetiolata* appears to be endemic to Tanzania, specifically to the Uzondo Plateau in Mpanda District, where it was collected at an elevation of 1,600 m (Fig 3). It inhabits seasonally inundated *Loudetia* Hochst. ex Steud. grasslands, growing on shallow sandy-peaty soil over sandstone rocks.

*Etymology.* The epithet refers to the length of the petiole which does not exceed 3 mm.

*Conservation notes.* *Acalypha brevipetiolata* is currently known from a single collection, and its population status remains unknown. We estimate its AOO at 4 km$^2$, while its EOO cannot be calculated due to the lack of data. The Uzondo Plateau remains largely undisturbed, but it is traversed by a recently-built road between Uvinza and Mpanda towns [22]. Plans to widen this road could increase habitat fragmentation and impact the species' habitat. Additionally, fires and logging represent potential long-term risks to the grassland ecosystem. Based on the available data, we preliminarily assess *A. brevipetiolata* as Vulnerable (VU) under IUCN criterion D2. However, if habitat degradation due to infrastructure development, fire, or logging accelerates, the species could qualify for reassessment as Endangered (EN) or even Critically Endangered (CR) in the future.

**3.1.2. *Acalypha bracteolata*. *Acalypha bracteolata*** I.Montero, Cardiel & P.Muñoz, *sp. nov.* [urn:lsid:ipni.org: names:77368650-1] Type: Tanzania, T6, Kilombero District: Udzungwa Mountains National Park, Mang'ula Village, 680 m, 7º44'38''S 36º53'14''E, 31 May 1999. *G. Massawe 282* (MO[MO-3625495], holo.; MA, iso.). Fig 4.

*Diagnosis.* *Acalypha bracteolata* is primarily characterised by its herbaceous habit, petioles up to 8 cm long, broadly ovate-lanceolate to elliptic-lanceolate leaf blades with pocket-shaped domatia at the base of the main veins, and terminal female inflorescences with bracts bearing simple and long glandular trichomes. The bract margins are irregularly dentate with 18–20 teeth, including a very prominent central tooth, and bracteoles up to 1.8 mm long.

*Description.* Perennial herb up to 2 m tall, monoecious. Branches pubescent with short, antrorsely curved, simple trichomes, glabrescent. Axillary buds perulate, perules 2, valvate, c. 0.8 mm long, triangular, subglabrous. Stipules conspicuous, up to 3.8 mm long, linear-lanceolate, with some simple trichomes up to 0.5 mm long and minute, sparse, glandular trichomes at margin. Petioles 3.2–7.5(–8) cm long, indumentum similar to that on young branches, glabrescent. Leaf blades (5.5–)6–11(–13) × 4–8(–8.5) cm, broadly ovate-lanceolate to elliptic-lanceolate, membranous; base rounded to

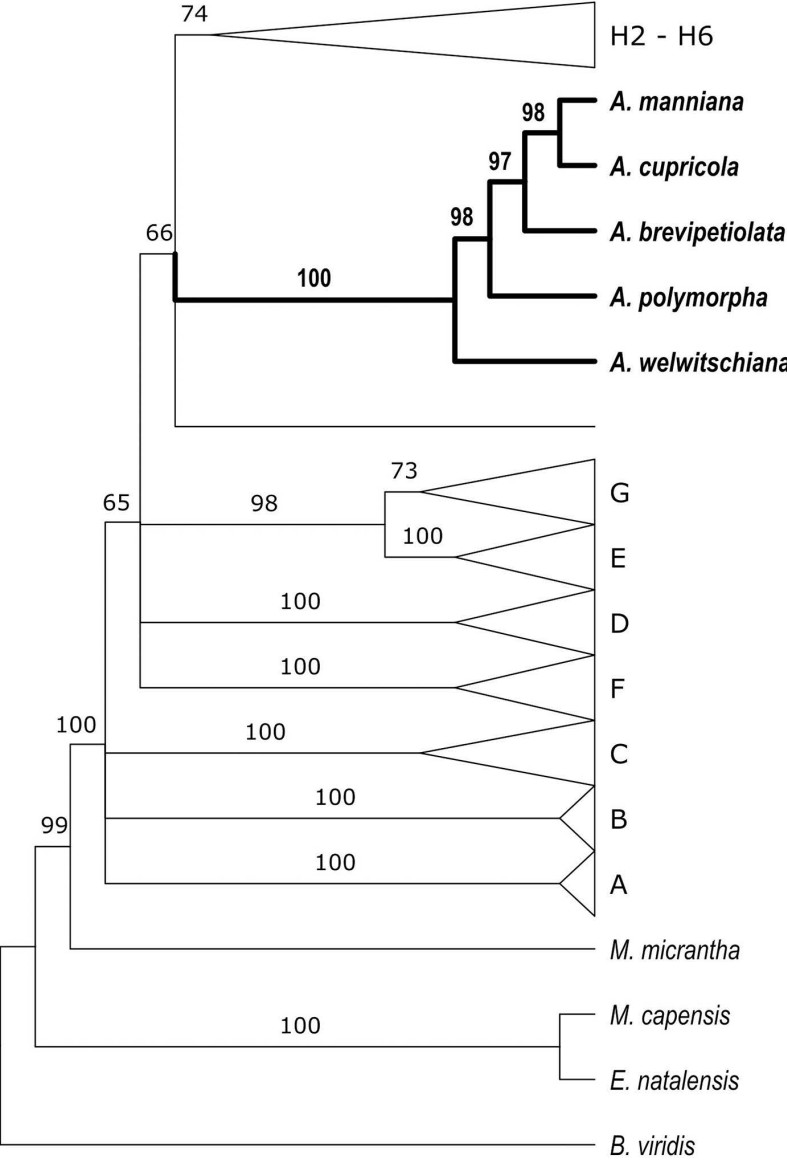

**Fig 2. nrITS phylogeny of *Acalypha* showing the position of *Acalypha brevipetiolata* within a clade alongside four other species.** The remaining clades, named following [Levin et al. (2022)], have been collapsed to enhance readability. The full phylogeny is available in S1 Fig.

subcordate; apex caudate, with a long acumen 1–2.3 cm long, mucronate; margins conspicuously serrate, teeth irregular, acute, usually with a minute, glandular trichome at apex; upper surface sparsely hairy with erect, simple trichomes up to 1 mm long; lower surface subglabrous, with appressed, simple trichomes mainly on veins; venation actinodromous, basal veins 3–5(–7), secondary veins 5–7 per side, with pocket-shaped domatia at the base on the main veins. <u>Inflorescences</u> unisexual, spiciform, male axillary, female terminal. <u>Male inflorescences</u> up to 14 cm long, densely flowered; peduncle to 1.3 cm long; flowers glomerate; bracts to 2 mm long, oblong, obtuse at apex, slightly hyaline at margin, ciliate with long, simple trichomes c. 1 mm long and some sparse, minute, glandular trichomes. <u>Female inflorescences</u> more or less densely flowered (with 40–50 bracts) up to 9.5 cm long; peduncle to 0.5 cm long, indumentum similar to that on young

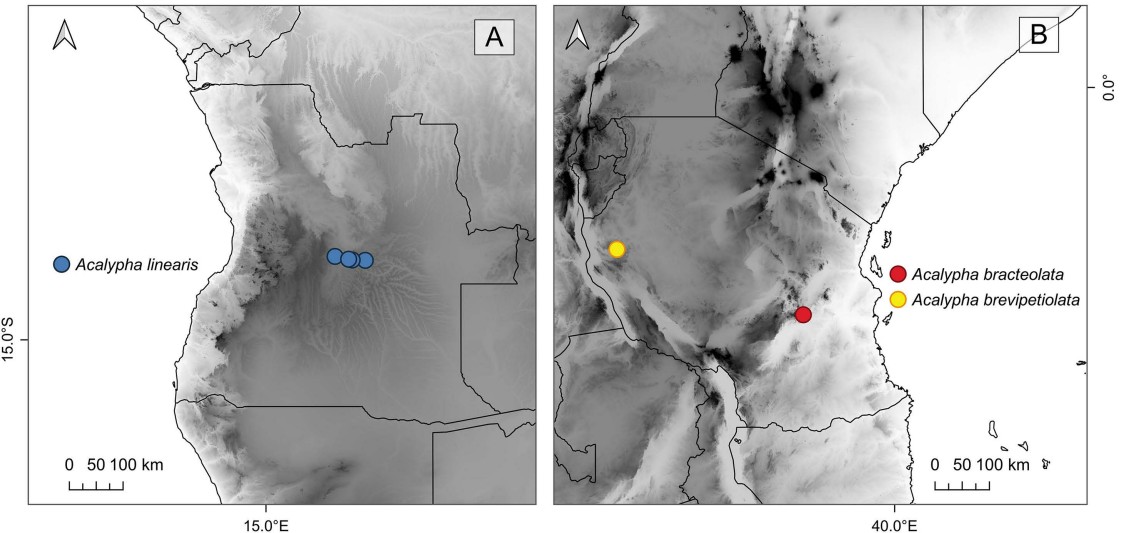

**Fig 3. Map of Angola (A) and Tanzania (B) showing the collection sites of the new species.** Blue, *Acalypha linearis* I.Montero, Cardiel & P.Muñoz (seven collections); red, *A.* bracteolata I.Montero, Cardiel & P.Muñoz (one collection); yellow, *A.* brevipetiolata I.Montero, Cardiel & P.Muñoz (one collection).

branches; bracts sessile, enlarging in fruit up to 6.2 × 4 mm, sparsely hairy with simple, short trichomes up to 0.5 mm long and long, glandular trichomes up to 0.7 mm long, mainly at margin and upper surface; margin irregularly dentate, teeth 18–20, triangular-lanceolate, acute, up to 1.5 mm long, with a very prominent central tooth up to 2.5 mm long; bracteoles up to 1.8 mm long, narrowly triangular to linear-lanceolate, with minute sessile glands at margin. Male flower with pedicel up to 0.8 mm long, glabrous; buds c. 0.6 mm diameter, glabrous. Female flower one per bract, sessile; sepals 3, up to 1.5 mm long, slightly connate at base, ovate-lanceolate, ciliate, with simple trichomes c. 0.3 mm long and some sparse sessile, glandular trichomes; ovary c. 0.7 mm diameter, 3-lobed, densely pubescent with simple trichomes up to 0.3 mm long; styles 3, up to 3.5 mm long, distinct, each divided into 8–10 slender branches, with some simple trichomes up to 0.4 mm long at rachis. Capsules present but immature. Seeds unknown.

*Distribution and habitat.* *Acalypha bracteolata* appears to be endemic to Tanzania, specifically to Kilombero District within Udzungwa Mountains National Park, where it was collected at an altitude of 680 m (Fig 3). It inhabits lowland rainforest and miombo woodland.

*Conservation notes.* *Acalypha bracteolata* is currently known from a single collection in Udzungwa Mountains National Park, and its population status remains unknown. We estimate its AOO at 4 km², while its EOO cannot be calculated due to the limited distribution data. Udzungwa Mountains National Park has been a protected area since 1992 and is recognised for its exceptional biodiversity and high levels of endemism [23]. However, the region has experienced extensive environmental degradation, with approximately 76% of its primary vegetation lost due to logging and grazing over the past decades. While these pressures have declined since the 1970s, the spread of invasive species now poses the primary ecological threat [24].

Based on its highly restricted distribution and potential vulnerability to ongoing habitat changes, we preliminarily assess *A. bracteolata* as Vulnerable (VU) under IUCN criterion D2. However, if invasive species continue to degrade its habitat, the species could face further reassessment as Endangered (EN) or even Critically Endangered (CR) in the future.

*Etymology.* The epithet refers to the presence of conspicuous bracteoles at the base of the female bracts, a feature that is relatively common in *Acalypha* species from Madagascar but infrequent in continental African species.

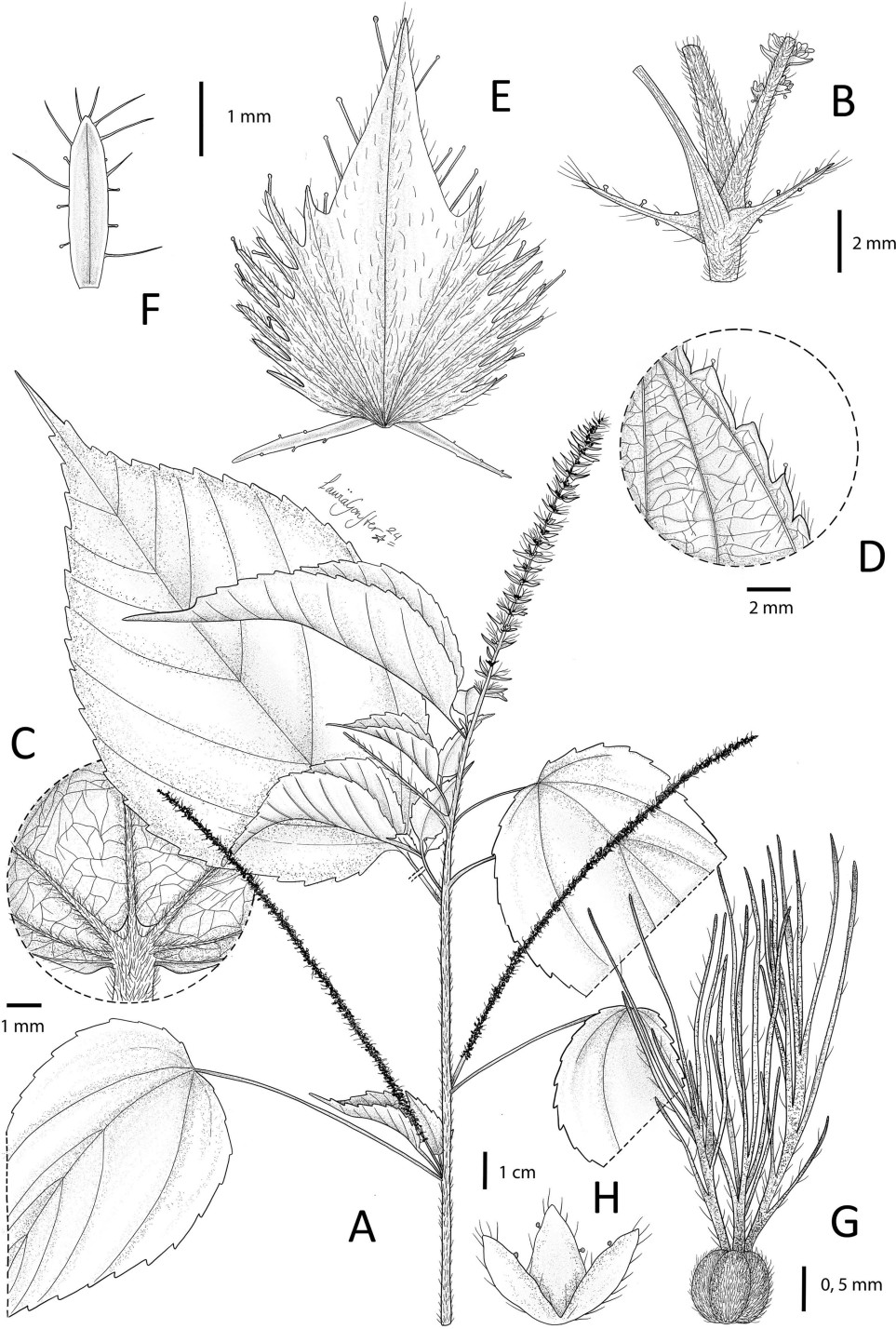

**Fig 4.  *Acalypha bracteolata* I. Montero, Cardiel & P.Muñoz.**  A. Habit. B. Detail of node, stipules, petiole base and male inflorescence base. C. Detail of base of lower leaf surface. D. Detail of leaf margin. E. Mature female bract. F. Mature male bract G. Ovary and styles. H. Calyx of the female flower. Based on *G. Massawe 282*. Illustration by Laura González Hernández.

### 3.1.3. *Acalypha linearis*.

*Acalypha linearis* I.Montero, Cardiel & P.Muñoz, *sp. nov*. [urn:lsid:ipni.org: names:77368651-1] Type: Angola, Bié Province, Cuemba, 17 km from Munhango, 12º08'43''S 18º05'11''E, 19 Sep 1965. *R. Mendes dos Santos 1881* (LISC[LISC052457], holo.). Fig 5.

*Diagnosis. Acalypha linearis* is primarily characterised by its numerous unbranched deeply striate stems arising from a woody rootstock; sessile, linear to aciculiform leaves with entirely or partially revolute margins; and terminal female inflorescences up to 3 cm long, with bracts densely pubescent with simple trichomes, and margins regularly dentate with 7−9 teeth.

*Description.* Perennial herb up to 35 cm tall, with many unbranched stems from a woody rootstock, monoecious or dioecious. Stems deeply striate, subglabrous, with some short, simple trichomes towards the apex. Stipules conspicuous, up to 2.5 mm long, narrowly triangular to linear-lanceolate, arched, pubescent with simple, short trichomes c. 0.6 mm long. Leaves sessile. Leaf blades (2.5−)3−8.5(−9) × 0.1−0.2(−0.3) cm, linear to aciculiform, chartaceous; apex acute, slightly callose; margins totally or partially revolute, mostly entire, denticulate towards the apex; upper and lower surface glabrous; venation pinnate, secondary veins inconspicuous, c. 10 per side. Inflorescences unisexual, spiciform, usually in different stems, male axillary, female terminal. Male inflorescences up to 3.5 cm long; peduncle up to 1.3 cm long, glabrous; flowers glomerate; bracts up to 1.2 mm long, linear to subspathulate, pubescent, with short, simple trichomes c. 0.5 mm long. Female inflorescences up to 3 cm long; peduncle up to 1.2 cm long, pubescent with some short, simple, slender trichomes c. 1 mm long, glabrescent; bracts 7−8, sessile, enlarging in fruit up to 5.5 × 7 mm, minutely papillose, densely pubescent with short, simple, trichomes c. 0.5 mm long; margin regularly dentate, teeth 7−9, triangular, acute, ½–⅓ of the total length; bracteoles absent. Male flowers with pedicel up to 1 mm long, sparsely hairy with short, simple trichomes c. 0.7 mm long; buds c. 0.8 mm diameter, sparsely hairy with simple, short, slender trichomes c. 0.7 mm long, mainly distally. Female flowers 1 per bract, sessile or subsessile; sepals 3, up to 1.2 mm long, distinct, oblong-lanceolate, ciliate with short, simple trichomes c. 0. 5 mm long; ovary c. 1 mm diameter, 3-lobed, densely hispid with simple, short trichomes; styles 3, up to 11 mm long, slightly connate at base, each with 8–15 short, simple branches, sparsely hairy with some, short, simple trichomes at rachis. Capsules up to 6 mm diameter, surface smooth, densely pubescent with short, simple trichomes c. 0.7 mm long, mainly towards the apex. Seeds c. 3 × 2.1 mm, pyriform, minutely foveolate.

*Additional specimens.* Angola. Bié Province: General-Machado [Camacupa], 1 km from Savinguila do Cuemba, 1100 m, [12º01'03''S 17º28'24.9''E], 4 Oct 1965. *B. Teixeira et al. 8971* (LISC052360, LISC052530); General-Machado, [Camacupa], Munhango, 1100 m, [12º09'28''S 18º33'20''E], 13 Sep 1965. *B. Teixeira et al. 8854* (LISC052361, LISC052530); Cuemba, riverbank of Cuiba river, 36 km from Decumba, [12º08'43''S 18º05'11''E], 27 Aug 1965. *R. Mendes dos Santos 1732* (LISC052475); Cuemba, [12º08'43''S 18º05'11''E], 13 Sep 1965. *R. Monteiro e Murta 1713* (LISC052522); Cuiba, in the Kalahari sands, [12º07'01''S 17º57'32''E], 27 Aug 1965. *R. Monteiro e Murta 1564* (LISC052525); Camacupa, Cuemba, Cuiva, 1200 m, [12º07'01''S 17º57'32''E], 27 Aug 1965. *B. Teixeira & R. Monteiro e Murta 8732* (LISC052529).

*Distribution and habitat. Acalypha linearis* appears to be endemic to Angola and is known only from Bié Province, where it occurs at an altitude of approximately 1,100 m (Fig 3). It inhabits the Miombo woodlands ecoregion [25], particularly in ecotonal areas characterised by shrub formations and sandy soils.

*Preliminary conservation status. Acalypha linearis* is currently known from seven collections, all made in Bié province, Angola, between August and September 1965. These collections originate from closely located sites, representing a single threat-based location. With no recent records available, the current population status remains unknown. We estimate the species' EOO at approximately 344 km², and its AOO at 16 km².

All collections were made outside Angola's protected areas network. *Acalypha linearis* occurs in woodlands and open forests, ecosystems that are increasingly threatened by agricultural expansion [26,27]. The ongoing conversion of these habitats may lead to a decline in habitat quality, potentially reducing the species' viability. Based on its restricted distribution, lack of recent records, and increasing habitat threats, we preliminarily assess *A. linearis* as Endangered (EN) under IUCN criteria B1ab(iii)+2ab(iii).

*Etymology.* The epithet refers to the characteristically sessile, linear leaves.

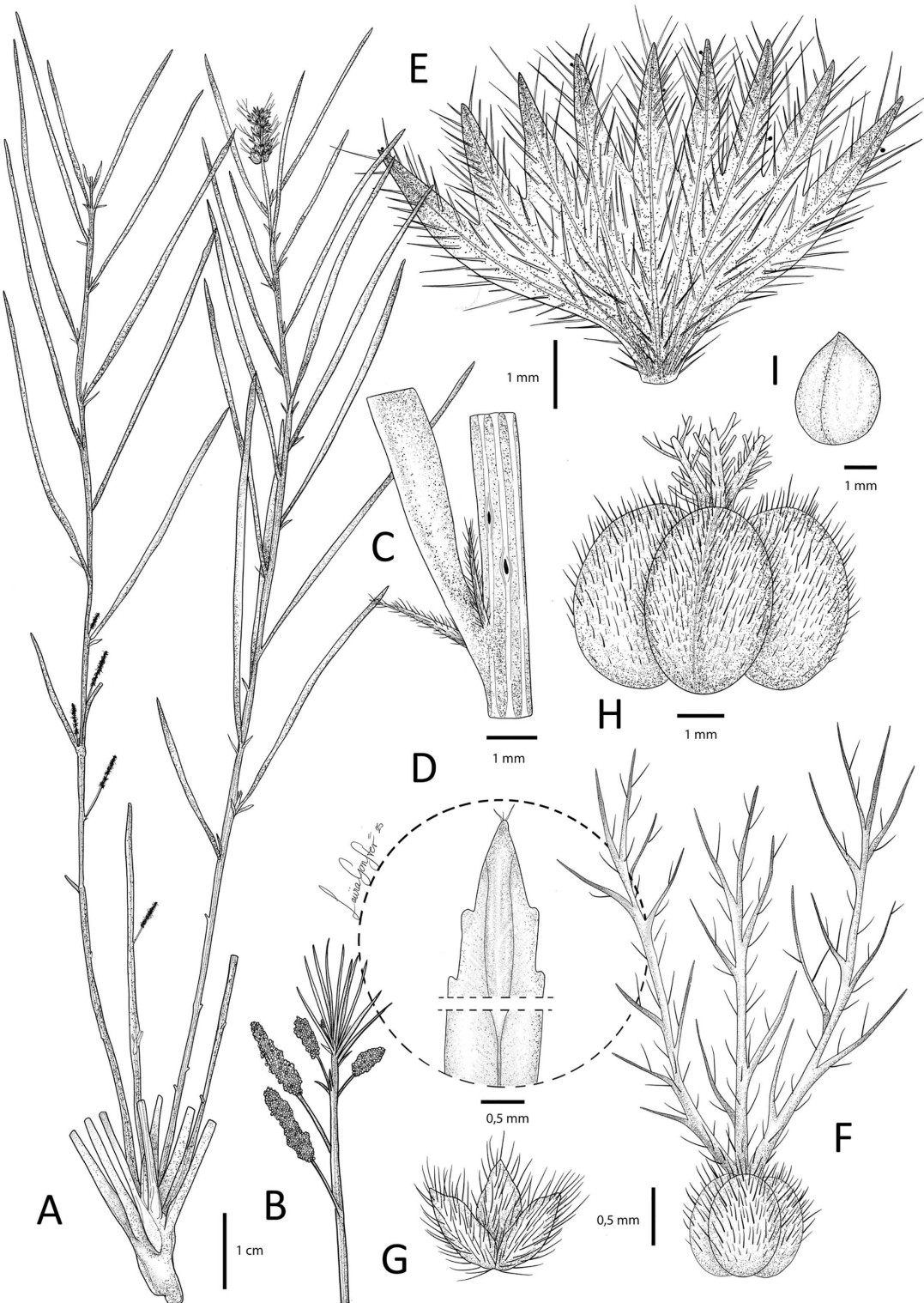

**Fig 5. *Acalypha linearis* I. Montero, Cardiel & P.Muñoz.** A. Habit. B. Detail of a branch with male inflorescences. C. Detail of node, stipules and leaf base. D. Detail of apex of lower leaf surface. E. Mature female bract. F. Ovary and styles. G. Calyx of the female flower. H. Capsule. I. Seed. Based on *R. Mendes dos Santos 1881*. Illustration by Laura González Hernández.

## 4. Discussion

The three new species described here belong to *Acalypha* subgenus *Acalypha*. Species in this subgenus can be easily recognised within *Acalypha* by their sessile or subsessile female flowers, calyx with 3 sepals, and generally accrescent, foliaceous bracts in fruit [11].

*Acalypha brevipetiolata* is part of a strongly supported but geographically widespread African clade that currently includes *A. cupricola*, *A. manniana*, *A. polymorpha*, and *A. welwitschiana* (Fig 2, Table 1). Additional species may belong to this clade as more sequencing is conducted. Of the currently included species, only *A. polymorpha* and *A. welwitschiana* have been recorded in Tanzania. *Acalypha cupricola* is endemic to the Democratic Republic of the Congo, while *A. manniana* is more widespread, occurring in neighbouring countries such as Uganda, Rwanda, and Burundi, though it has not been recorded in Tanzania. The phylogenetic affinity of *A. brevipetiolata* with these species suggests a broader African lineage that remains understudied.

Morphologically, *Acalypha brevipetiolata* bears a superficial resemblance to *A. linearifolia* Leandri, a species endemic to Madagascar's littoral forests and thickets [3]. However, beyond their occurrence in different habitats, they also differ in key diagnostic features of their habit (shrubby in *A. linearifolia* vs. herbaceous in *A. brevipetiolata*), female bracts (with stellate trichomes and entire to denticulate margins in *A. linearifolia* vs. with simple trichomes and deeply dentate margins in *A. brevipetiolata*), and capsules (papillose with stellate trichomes in *A. linearifolia* vs. smooth with simple trichomes in *A. brevipetiolata*). The absence of molecular data for *A. linearifolia* prevents direct phylogenetic comparison.

*Acalypha linearis*, the new species from Angola, resembles *A. brevipetiolata*, particularly in habit and leaf shape, but differs in other key characters already mentioned in the descriptions. Both species appear to be geographically restricted to opposite sides of the continent, with *A. linearis* likely occurring at slightly lower elevations. Both species are associated with fire-prone ecosystems characterised by marked seasonality and sandy soils [25] and —along with several other African *Acalypha* representatives— possess a woody rootstock, an adaptation commonly linked to fire resilience [28]. Given

**Table 1. Key morphological characters distinguishing species within the same clade as *Acalypha brevipetiolata*.**

| Characters | *A. brevipetiolata* | *A. cupricola* | *A. manniana* | *A. polymorpha* | *A. welwitschiana* |
|---|---|---|---|---|---|
| **Habit** | perennial herb | perennial herb | shrub | perennial herb | shrublet |
| **Stipule length** | 0.5 mm long | 1–2 mm long | 3 mm long | 1–3 mm long | 1.5–3 mm long, |
| **Stipule shape** | triangular | linear | triangular-lanceolate | linear-subulate | linear-subulate |
| **Petiole length** | 1−3 mm | 0.5–4 mm | 2−3 cm | 0.1–1 cm | 0.2–5 cm |
| **Leaf blade shape** | linear to linear-lanceolate | narrowly ovate | ovate-lanceolate | suborbicular to ovate, obovate, elliptic, oblong, narrowly oblanceolate or lanceolate | elliptic-ovate to lanceolate |
| **Leaf blade size** | 2.5−4 × 0.2−0.4 cm | 4–6 × 0.8–1.2 cm | 6.5–8 × 4–4.5 cm | 1–10 × 0.5–2.5 cm | 1–13 × 0.5–7.5 cm |
| **Leaf margin** | entire to sparsely denticulate | serrulate | serrate | serrate or crenate-serrate | crenate to crenate-serrate |
| **Main nerves** | 1 | 1 | 5 | 3–5(7) | 5 |
| **Inflorescences** | unisexual | bisexual and unisexual | unisexual | unisexual and sometimes bisexual | unisexual |
| **Female inflorescence** | axillary | axillary | axillary and terminal | axillary and terminal | axillary and terminal |
| **Developed female bract margin** | irregularly dentate, teeth 8–9 | irregularly dentate, teeth 14–20 | dentate-laciniate, teeth 9–13 | dentate-laciniate, teeth 8–10 | dentate-laciniate, teeth 15–25 |
| **Developed female bract size** | 5 × 5 mm | 7–10 × 10–12 mm | 9 × 9 mm | 1 × 1.5 cm | 0.6–1 × 1–1.3 cm |
| **Developed female bract indument** | pubescent, simple and glandular trichomes | densely puberulent, only simple trichomes | pubescent, simple and glandular trichomes | pubescent, only simple trichomes | pubescent, only simple trichomes |

that this trait is also present in other members of the *A. brevipetiolata* clade, it would not be surprising if molecular analyses reveal a close relationship between *A. linearis* and its sequenced counterpart. These findings suggest that certain *Acalypha* lineages have undergone ecological adaptations to seasonally dry and fire-influenced habitats across Africa.

Finally, **Acalypha bracteolata**, the other Tanzanian species described here, is morphologically similar to *A. ornata* Hochst. ex A.Rich., a widespread African species, but differs in key traits, including their habit (herbaceous in *A. bracteolata* vs. shrubby in *A. ornata*), female inflorescences (up to 9.5 cm long in *A. bracteolata* vs. up to 15 cm long in *A. ornata*) and developed female bracts (up to 6.2 × 4 mm with conspicuous bracteoles in *A. bracteolata* vs. up to 12 × 11 mm and without bracteoles in *A. ornata*). However, due to the limited knowledge of the genus in Africa, its phylogenetic placement remains uncertain without molecular data.

This highlights a broader challenge: our understanding of African *Acalypha* is constrained by the overall lack of both morphological and molecular studies. As a result, it is often difficult to determine whether observed similarities reflect shared ancestry or convergent evolution in similar ecological settings.

Of the three species, we obtained high-quality DNA and successfully sequenced *Acalypha brevipetiolata*, allowing its placement in a phylogenetic framework. The evolutionary relationships of *A. bracteolata* and *A. linearis* remain uncertain due to the lack of molecular data, highlighting the need for further sequencing efforts. Future phylogenetic studies incorporating additional African taxa will be essential to resolve their placement within the genus and clarify species boundaries.

### 4.1. Conservation

The two new Tanzanian species—*A. brevipetiolata* and *A. bracteolata*—are currently known from only a single collection each, and their population status remains unknown. This represents a major limitation in assessing their conservation status. Similarly, *A. linearis* is known from seven collections but they were all made in 1965 in the same Angolan region, thus representing a single threat-based locality. To the best of our knowledge, this species has not been recorded in over sixty years, leaving its current population status uncertain. Given the rarity of these species and their restricted known distributions, urgent field surveys are needed to locate additional populations, assess potential threats, and evaluate their conservation status.

### 4.2. Taxonomic implications

As noted by [8], species descriptions for mainland Africa have been rare in recent decades, with only two new *Acalypha* species published in the last 30 years [29] —compared to 30 from the Americas, 7 from Asia, and 17 from Madagascar and surrounding islands over the same period. The gap is even more striking for Tanzania and Angola: *A. gillmanii* Radcl.-Sm., the last species described from Tanzanian material, was published in 1975, while no new species have been described from Angolan specimens since Pax and Hoffman's *Das Pflanzenreich* a century ago [30]. In this context, our description of three new species represents the first formal addition to these countries' *Acalypha* flora in decades. This underscores the need for continued botanical exploration and taxonomic revision in the region.

Ongoing taxonomic work, including herbarium research, molecular studies and field surveys, is critical to delimiting species boundaries and elucidating evolutionary relationships within *Acalypha*. In-person visits to herbaria —both within Tanzania and Angola and in former colonial institutions where their historical collections are housed— along with targeted specimen loans remain essential to our work. These efforts will contribute to a more comprehensive understanding of *Acalypha* diversity and biogeography in Africa, a region where we anticipate many *Acalypha* species have yet to be described.

### Supporting information

**S1 Fig. Extended nrITS phylogeny of *Acalypha* showing the position of *Acalypha brevipetiolata*, in red, forming a clade with four other species: *A. manniana*, *A. cupricola*, *A. polymorpha* and *A. welwitschiana*.**
(SVG)

**S1 Table. Extraction protocol using the CTAB method, adapted from Doyle (1991).**
(DOCX)

**S1 File. Phylogenetic analysis files.**
(ZIP)

## Acknowledgments

The authors would like to acknowledge the herbarium curators at BR, K, L, LISC, LISU, MA, MO, P, and WAG for access to their collections.

## Author contributions

**Conceptualization:** Iris Montero-Muñoz, José María Cardiel, Pablo Muñoz-Rodríguez.

**Data curation:** Iris Montero-Muñoz, José María Cardiel, Pablo Muñoz-Rodríguez.

**Formal analysis:** Lucía Villaescusa-González.

**Funding acquisition:** Pablo Muñoz-Rodríguez.

**Investigation:** Iris Montero-Muñoz, José María Cardiel, Pablo Muñoz-Rodríguez.

**Project administration:** Pablo Muñoz-Rodríguez.

**Validation:** Iris Montero-Muñoz, José María Cardiel, Pablo Muñoz-Rodríguez.

**Visualization:** Iris Montero-Muñoz, José María Cardiel, Pablo Muñoz-Rodríguez.

**Writing – original draft:** Iris Montero-Muñoz, José María Cardiel, Pablo Muñoz-Rodríguez.

**Writing – review & editing:** Iris Montero-Muñoz, José María Cardiel, Pablo Muñoz-Rodríguez.

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
