## [Decision Letter · Decision Letter 0]

17 Jul 2025

PONE-D-25-22700Three new species of Acalypha L. (Euphorbiaceae, Acalyphoideae) from Tanzania and Angola and their conservation statusPLOS ONE

Dear Dr. Muñoz Rodríguez,

Thank you for submitting your manuscript to PLOS ONE. After careful consideration, we feel that it has merit but does not fully meet PLOS ONE’s publication criteria as it currently stands. Therefore, we invite you to submit a revised version of the manuscript that addresses the points raised during the review process.

We look forward to receiving your revised manuscript.

Kind regards,

Andrea Mastinu

Academic Editor

PLOS ONE

Journal Requirements:

2. Please take this opportunity to be sure you have met all of our guidelines for new species. When publishing papers that describe a new botanical taxon, PLOS aims to comply with the requirements of the International Code of Nomenclature for algae, fungi, and plants (ICN). In association with the International Plant Names Index (IPNI), the following guidelines for publication in an online-only journal have been agreed such that any scientific botanical name published by us is considered effectively published under the rules of the Code. Please note that these guidelines differ from those for zoological nomenclature, and apply only to seed plants, ferns, and lycophytes.

Effective January 2012, "the description or diagnosis required for valid publication of the name of a new taxon" can be in either Latin or English. This does not affect the requirements for scientific names, which are still to be Latin.

Also effective January 2012, the electronic PDF represents a published work according to the ICN for algae, fungi, and plants. Therefore the new names contained in the electronic publication of a PLOS ONE article are effectively published under that Code from the electronic edition alone, so there is no longer any need to provide printed copies.

For proper registration of the new taxon, we require two specific statements to be included in your manuscript.

a.        In the Methods section, include a sub-section called "Nomenclature" using the following wording:

The electronic version of this article in Portable Document Format (PDF) in a work with an ISSN or ISBN will represent a published work according to the International Code of Nomenclature for algae, fungi, and plants, and hence the new names contained in the electronic publication of a PLOS ONE article are effectively published under that Code from the electronic edition alone, so there is no longer any need to provide printed copies.

In addition, new names contained in this work have been submitted to IPNI, from where they will be made available to the Global Names Index. The IPNI LSIDs can be resolved and the associated information viewed through any standard web browser by appending the LSID contained in this publication to the prefix http://ipni.org/. The online version of this work is archived and available from the following digital repositories: [INSERT NAMES OF DIGITAL REPOSITORIES WHERE ACCEPTED MANUSCRIPT WILL BE SUBMITTED (PubMed Central, LOCKSS etc)].

All PLOS ONE articles are deposited in PubMed Central and LOCKSS. If your institute, or those of your co-authors, has its own repository, we recommend that you also deposit the published online article there and include the name in your article.

b.        In the Results section, the globally unique identifier (GUID), currently in the form of a Life Science Identifier (LSID), should be listed under the new species name, for example:

Solanum aspersum S.Knapp, sp. nov. [urn:lsid:ipni.org:names:77103633-1] Type: Colombia. Putumayo: vertiente oriental de la Cordillera, entre Sachamates y San Francisco de Sibundoy, 1600-1750 m, 30 Dec 1940, J. Cuatrecasas 11471 (holotype, COL; isotypes, F [F-1335119], US [US-1799731]).

PLOS ONE staff will contact IPNI to obtain the GUID (LSID) after your manuscript is accepted for publication, and this information will then be added to the manuscript at that time. You may indicate where the number or numbers will be added using XXXXX or an equivalent placeholder.

A complete explanation of our guidelines for publishing new species can be found on our website: http://www.plosone.org/static/guidelines#botanical

“The authors are grateful for the financial support provided by the research project PID2022-139634NA-I00M funded by MCIN/AEI/10.13039/501100011033/FEDER, UE. PMR was supported by a Ramón y Cajal research contract, grant ref. RYC2021-032489-I, and IMM by a Juan de la Cierva – Formación-2021, grant ref. FJC2021-046607-I, funded by MCIN/AEI/ 10.13039/501100011033 and the European Union NextGenerationEU/PRTR during the work on this project.”

4. In the online submission form, you indicated that your data will be submitted to a repository upon acceptance.  We strongly recommend all authors deposit their data before acceptance, as the process can be lengthy and hold up publication timelines. Please note that, though access restrictions are acceptable now, your entire minimal  dataset will need to be made freely accessible if your manuscript is accepted for publication. This policy applies to all data except where public deposition would breach compliance with the protocol approved by your research ethics board. If you are unable to adhere to our open data policy, please kindly revise your statement to explain your reasoning and we will seek the editor's input on an exemption.

“The authors are grateful for the financial support provided by the research project PID2022-139634NA-I00M funded by MCIN/AEI/10.13039/501100011033/FEDER, UE. PMR was supported by a Ramón y Cajal research contract, grant ref. RYC2021-032489-I, and IMM by a Juan de la Cierva – Formación-2021, grant ref. FJC2021-046607-I, funded by MCIN/AEI/ 10.13039/501100011033 and the European Union NextGenerationEU/PRTR during the work on this project.”

" The authors are grateful for the financial support provided by the research project PID2022-139634NA-I00M funded by MCIN/AEI/10.13039/501100011033/FEDER, UE. PMR was supported by a Ramón y Cajal research contract, grant ref. RYC2021-032489-I, and IMM by a Juan de la Cierva – Formación-2021, grant ref. FJC2021-046607-I, funded by MCIN/AEI/ 10.13039/5011000”

6. Please include your tables as part of your main manuscript and remove the individual files. Please note that supplementary tables (should remain/ be uploaded) as separate "supporting information" files.

7. Please include captions for your Supporting Information files at the end of your manuscript, and update any in-text citations to match accordingly. Please see our Supporting Information guidelines for more information: http://journals.plos.org/plosone/s/supporting-information .

8. We note that Figure 3 in your submission contain [map/satellite] images which may be copyrighted. All PLOS content is published under the Creative Commons Attribution License (CC BY 4.0), which means that the manuscript, images, and Supporting Information files will be freely available online, and any third party is permitted to access, download, copy, distribute, and use these materials in any way, even commercially, with proper attribution. For these reasons, we cannot publish previously copyrighted maps or satellite images created using proprietary data, such as Google software (Google Maps, Street View, and Earth). For more information, see our copyright guidelines: http://journals.plos.org/plosone/s/licenses-and-copyright.

a. You may seek permission from the original copyright holder of Figure 3 to publish the content specifically under the CC BY 4.0 license. 

Reviewers' comments:

Reviewer's Responses to Questions

**Comments to the Author**

1. Is the manuscript technically sound, and do the data support the conclusions?

Reviewer #1: Yes

2. Has the statistical analysis been performed appropriately and rigorously? 

Reviewer #1: Yes

3. Have the authors made all data underlying the findings in their manuscript fully available?

Reviewer #1: Yes

4. Is the manuscript presented in an intelligible fashion and written in standard English?

Reviewer #1: Yes

5. Review Comments to the Author

Reviewer #1: The manuscript describes three new species of Acalypha from continental Africa, resulting from an extensive study of the group. The study is relevant as it sheds light on regions where few taxonomic studies have been conducted on the local flora. Overall, the manuscript is well written, with only a few minor corrections and typos, which have been annotated in the revised version.

Although the text provides context regarding the limited available information on African Acalypha species, it does not discuss the infrageneric classification, nor whether it is considered obsolete or of limited utility.

6. PLOS authors have the option to publish the peer review history of their article (what does this mean? ). If published, this will include your full peer review and any attached files.

**Do you want your identity to be public for this peer review?** For information about this choice, including consent withdrawal, please see our Privacy Policy .

Reviewer #1: **Yes: ** Jose Floriano Barêa Pastore

---

## [Author Response · Author response to Decision Letter 1]

13 Aug 2025

Dear Editor,

Thank you for considering our manuscript for publication in PLOS One. We submit a revised version of the manuscript addressing all reviewer’s suggestions, and provide comments to two reviewer’s comments and to your editorial comments in the rebuttal letter attached.

Best,

Pablo

---

## [Decision Letter · Decision Letter 1]

2 Sep 2025

Three new species of Acalypha L. (Euphorbiaceae, Acalyphoideae) from Tanzania and Angola and their conservation status

PONE-D-25-22700R1

Dear Dr. Muñoz Rodríguez,

We’re pleased to inform you that your manuscript has been judged scientifically suitable for publication and will be formally accepted for publication once it meets all outstanding technical requirements.

Kind regards,

Andrea Mastinu

Academic Editor

PLOS ONE

Additional Editor Comments (optional):

Reviewer #1:

Reviewers' comments:

Reviewer's Responses to Questions

**Comments to the Author**

1. If the authors have adequately addressed your comments raised in a previous round of review and you feel that this manuscript is now acceptable for publication, you may indicate that here to bypass the “Comments to the Author” section, enter your conflict of interest statement in the “Confidential to Editor” section, and submit your "Accept" recommendation.

Reviewer #1: All comments have been addressed

2. Is the manuscript technically sound, and do the data support the conclusions?

Reviewer #1: Yes

3. Has the statistical analysis been performed appropriately and rigorously? 

Reviewer #1: N/A

4. Have the authors made all data underlying the findings in their manuscript fully available?

Reviewer #1: Yes

5. Is the manuscript presented in an intelligible fashion and written in standard English?

Reviewer #1: Yes

6. Review Comments to the Author

Reviewer #1: The authors have adequately addressed all the comments and concerns raised in the previous round. I have no competing interests and no further ethical or publication concerns. I recommend the manuscript for publication.

7. PLOS authors have the option to publish the peer review history of their article (what does this mean? ). If published, this will include your full peer review and any attached files.

**Do you want your identity to be public for this peer review?** For information about this choice, including consent withdrawal, please see our Privacy Policy .

Reviewer #1: **Yes: ** Jose Floriano Barêa Pastore

---

## [Editor Report · Acceptance letter]

PONE-D-25-22700R1

PLOS ONE

Dear Dr. Muñoz-Rodríguez,

I'm pleased to inform you that your manuscript has been deemed suitable for publication in PLOS ONE. Congratulations! Your manuscript is now being handed over to our production team.

Kind regards,

on behalf of

Dr. Andrea Mastinu

Academic Editor

PLOS ONE